# Mammary Development in Gilts at One Week Postnatal Is Related to Plasma Lysine Concentration at 24 h after Birth, but Not Colostrum Dose

**DOI:** 10.3390/ani11102867

**Published:** 2021-09-30

**Authors:** Bryce Bitsie, Erin Kay Ison, Leah Parker Jenkins, Rebecca Klopp, Conor McCabe, Kayla Mills, Griffin Nicholls, Andrew Richards, Larissa Shirley, Kelsey Teeple, Allan P. Schinckel, Angela Kwon, Kara R. Stewart, Amber Jannasch, Aridany Suarez-Trujillo, Theresa M. Casey

**Affiliations:** 1Department of Animal Sciences, Purdue University, West Lafayette, IN 47907, USA; bbitsie@purdue.edu (B.B.); eison@purdue.edu (E.K.I.); jenki155@purdue.edu (L.P.J.); rklopp@purdue.edu (R.K.); cjmccabe@ucdavis.edu (C.M.); mills34@purdue.edu (K.M.); gnicholl@purdue.edu (G.N.); richa552@purdue.edu (A.R.); lshirle@purdue.edu (L.S.); kteeple@purdue.edu (K.T.); aschinck@purdue.edu (A.P.S.); kwon123@purdue.edu (A.K.); krstewart@purdue.edu (K.R.S.); asuarezt@berry.edu (A.S.-T.); 2Metabolite Profiling Facility Bindley Bioscience Center, Purdue University, West Lafayette, IN 47907, USA; hopfas@purdue.edu

**Keywords:** colostrum, gilt, mammary development, lysine, perinatal nutrition

## Abstract

**Simple Summary:**

A relationship exists between a female’s early nutritional environment and her ability to produce milk when she lactates as an adult. Colostrum is the first milk available to neonates after birth. We hypothesized that differing levels of colostrum stimulate differences in very early mammary development. Despite differences in weight at 24 h and 7 days, mammary morphological development and DNA content was not found to be different between gilts fed a high versus low dose of colostrum. The rate of mammary gland protein and DNA synthesis over the first week was not different between the groups. Circulating levels of amino acids were determined after 24 h of colostrum feeding, and levels of circulating lysine were found to be related to average daily gain and mammary DNA synthetic rate. Moreover, the level of lysine was related to a lower ratio of DNA to protein synthesis, suggesting that higher lysine favored cell division versus differentiation (by leaving the cell cycle). Further studies are needed in this area.

**Abstract:**

Perinatal nutrition affects future milk production. The number of mammary epithelial cells affect milk production capacity. Therefore, it was hypothesized that the level of colostrum intake affects the proliferation rate and the total number of mammary epithelial cells in the gland. The ratio of newly synthesized protein to newly synthesized DNA reflects the relative amount of cellular differentiation to cell division. The study objective was to determine the relationship between the level of colostrum intake and 24 h-level of circulating amino acid, glucose and insulin with mammary parenchyma histological features, cell division and protein synthesis over the first week postnatal. One of two standardized doses of a homogenate colostrum sample, 10% (*n* = 8) and 20% (*n* = 8) of birth bodyweight, was fed to gilts over the first 24 h postnatal. Gilts were administered deuterium oxide immediately after birth and daily to label newly synthesized DNA and proteins. Gilts were euthanized on postnatal day seven, and DNA and protein were isolated from mammary parenchyma. DNA and protein fractional synthesis (f) and fractional synthetic rate (FSR) were calculated using mass isotopomer distribution analysis. The ratio of protein f and FSR to DNA f and FSR were calculated and used to indicate the relative amounts of differentiation to cell division. Mammary morphological development was also analyzed by measuring the parenchymal epithelial area and the stromal and epithelial proliferation index on postnatal day seven. Colostrum dose was not related to any of the variables used to evaluate mammary development. However, plasma lysine levels at 24 h postnatal were positively related to average daily gain (ADG; *r* = 0.54, *p* = 0.05), DNA f (*r* = 0.57; *p* = 0.03) and DNA FSR (*r* = 0.57; *p* = 0.03) in mammary parenchyma. Plasma lysine was inversely related to the ratio of protein to DNA f and FSR (*r* = −0.56; *p* = 0.04). ADG was related to the parenchymal epithelial area and DNA and protein f and FSR (*p* < 0.05). These relationships support the idea that the nutritional environment affects early mammary development and that higher lysine levels in the perinatal period favored a greater degree of cell division versus differentiation in mammary of neonatal pigs and thus, warrant further investigations.

## 1. Introduction

The first days postnatal are a critical period of metabolic-nutritional programming in pigs. Of particular interest to this study is the relationship between early nutritional environment and future lactation performance. In swine, greater colostrum intake by gilts was related to earlier puberty and better lactation performance (as sows) than the low colostrum intake counterparts [1]. Studies in sheep showed the nutrition of ewes during pregnancy affected the yield and composition of milk produced by the offspring [2,3]. The preweaning growth rate of heifers was positively related to their milk production as cows [4]. Heifer dairy calves fed two liters of colostrum produced less milk in their first and second lactations than calves who were fed four liters [5]. The lower milk production was related to greater rates of morbidity and lower body weights of calves. Heifer calves fed restricted versus ad libitum intake of milk replacer had less mammary gland mass, mammary parenchyma, fat pad mass, and lower expansion of epithelium into the adjacent stromal tissue [6].

The number of mammary epithelial cells is highly correlated to milk production [7]. Mammary cell number is established during the development of the gland, which begins in utero [8]. The development of mammary glands, from birth to the peripubertal period, in pigs is characterized by ductal elongation and formation of a lumen [9]. The pattern of ductal branching in swine is similar to the human breast, which is characterized by terminal ductal lobular units (TDLU). Prior to the onset of puberty, mammary branching and organization increases in complexity from primarily TDLU-1 to TDLU-2 through the proliferation of lobular buds and ductal elongation.

Knowing that nutrition in early life impacts future milk production and that milk production is determined by the number of mammary epithelial cells led us to hypothesize that the level of colostrum intake during the first 24 h postnatal affects the proliferation rate of mammary epithelial cells in the gland, which in the long term would affect the number of milk-producing epithelial cells. Stem and progenitor cells populate tissues by asymmetric cell division [10]. The proliferation and self-renewal of stem-progenitor cells are balanced in the tissue by daughter cells exiting the cell cycle and beginning the differentiation process. As cells differentiate, they become more specialized, and this specialization is marked by protein synthesis [11]. In the gland, at any point in time, there is an array of cellular states as cells progress towards differentiated states [12]. Accounting for these parameters, we posited that the ratio of newly synthesized protein to newly synthesized DNA could be used as an indicator of the relative amount of cellular differentiation to cell division. 

The overall goal of this study was to test this hypothesis and determine the relationship between the level of colostrum intake and 24 h level of circulating amino acid, glucose and insulin with mammary parenchyma histological features, as well as cell division and protein synthesis over the first postnatal week. For the described studies, one of two standardized doses of a homogenate colostrum sample, 10% and 20% of birth bodyweight, was fed to study animals [13]. Gilts were given a bolus of heavy water (deuterium oxide) immediately after birth and daily for seven days to label newly synthesized DNA and protein over this time period. DNA and protein were isolated from mammary parenchymal tissue. DNA and protein fractional synthesis (f) and fractional synthetic rate (fraction per day, FSR) were calculated using mass isotopomer distribution analysis (MIDA) [14,15,16,17]. The ratio of protein f and FSR to DNA f and FSR were calculated and used to indicate the relative amount of differentiation to cell division. Mammary morphological development was also analyzed by measuring the epithelial area of parenchymal tissue and the stromal and epithelial proliferation index on postnatal day seven.

## 2. Materials and Methods

### 2.1. Animals and Study Design

Animals and Study Design (Figure 1). Prior to the start of this study, all animal procedures were reviewed and approved by the Institutional Animal Care and Use Committee (Protocol # 1907001920). Animals used for the study were born at the Purdue University Animal Sciences Research and Education Center Swine Farm to eight York (×) Landrace multiparous (3.25 ± 1.67 parities) sows bred to terminal sire Duroc boars. Sixteen animals (*n* = 16) used for the investigations were part of a larger study [13]. Animals were selected at birth from across eight different litters (*n* = 2 gilts/litter; birth litter size was 12.3 ± 1.3 live piglets/sow). At birth, gilts were towel-dried, weighed and assigned to one of two study treatments (1 gilt/litter/treatment); colostrum fed at a rate of 24 h intake of 10% of birth body weight (BW; COL10) or 20% of BW (COL20). Gilts that were selected weighed between 1.2 and 1.8 kg and treatments were matched across litters by weight. All gilts were bottle-fed with a pooled colostrum sample every 2 h from birth until 24 h of age. Between feedings, they were returned to a nursery area, which was maintained at 40 °C. Nursery temperature was chosen to prevent any risk of hyperthermia at this age, as this temperature is the same as the offsprings’ in utero environment.

A blood sample was collected from each gilt at 24 h postnatal, via jugular venipuncture, using a 22 gauge × 2.5 cm needle into a 2 mL potassium-EDTA coated vacutainer tube (BD367841, BD, Franklin Lakes, NJ, USA). At 24 h, body weight, body temperature and blood samples for plasma preparation were collected from each gilt to measure blood glucose, insulin, total protein and amino acid. Plasma was prepared by immediately centrifuging blood at 2000× *g* for 15 min (E8 Centrifuge, LW Scientific Inc., Lawrenceville, GA, USA).

Gilts were returned to their birth dam to be nursed for the remaining days of the study. Nursing litter size was standardized to 12–14 piglets per sow. Body weights were recorded daily for each gilt. Gilts in the COL10 and COL20 groups were euthanized on day seven postnatal. Animals were euthanized using CO_2_ inhalation, and then mammary tissue was collected by doing an incision longitudinally along both sides of the left mammary chain. Additionally, six gilts weighing between 1.2 and 1.8 kg were identified at birth, immediately euthanized and used to evaluate mammary morphology at birth.

The whole mammary chain was removed by dissecting through the subcutaneous tissue. For histology, a square of 1.5 × 1.5 cm of skin around the nipple was sectioned and placed in 10% buffered formalin. After 24 h, histology samples were transferred to 1X PBS. Mammary tissue used for MIDA analysis of DNA and protein synthesis was collected from the thoracic mammary glands, and mammary parenchyma was grossly dissected to remove muscle, skin and other tissue not associated with gland parenchyma, snap-frozen in liquid nitrogen and stored at −80 °C.

A sample size power analysis was performed prior to the start of the study. The power of the study with six animals per treatment with an alpha-error of 0.05, 1.5-fold difference between treatments and 0.25 standard deviation was 0.95. If the difference dropped to 1.4-fold, the power of the study was 0.8 with six animals. Since we anticipated the potential of loss of piglets, the study was begun with eight animals per treatment. Following tissue and plasma collection, all researchers were blinded to treatment during the experimental analysis portion of the study. The treatment groups were revealed for data compilation and statistical analysis on the effect of treatment. 

### 2.2. Colostrum Sample and Analysis

Approximately 50 mL of colostrum was collected from multiple sows (~250) over the course of 7 mo. Colostrum collection was done manually during active farrowing when oxytocin levels are naturally high. Following collection, colostrum was frozen and stored at −80 °C until the day prior to the start of the study. A homogenate-pooled sample was prepared following overnight thawing of colostrum at 4 °C. Piglets were fed this homogenate sample, and several aliquots were collected and stored at −80 °C for subsequent composition analysis. 

Colostrum composition was analyzed for percent fat, protein, and insulin concentration. Fat percentage was determined using the creamatocrit approach by centrifuging homogenate samples at 12,000× *g* for 10 min in a non-heparinized hematocrit tube (3 tubes per sample). Fat percentage was calculated as the ratio of the length of fat to total sample length measured with a caliper and then multiplied by 100.

The protein content of colostrum samples was measured using the Bradford Assay Kit (Pierce Coomassie Plus Assay Kit, Thermo Fisher Scientific, Waltham, MA, USA). Samples were diluted at 1:100 in phosphate buffer manufacturer’s instructions were followed. A plate spectrophotometer (Sparks 10M multimode microplate reader, Tecan) was used to analyze absorbance at 495 nm wavelength.

Colostrum composition was analyzed for percent fat, protein and insulin concentration. Fat percentage was determined using the creamatocrit approach by centrifuging homogenate samples at 12,000× *g* for 10 min in a non-heparinized hematocrit tube (3 tubes per sample). Fat percentage was calculated as the ratio of the length of fat to total sample length measured with a caliper then multiplied by 100. The protein content of colostrum samples was measured using a Bradford Assay Kit (Pierce Coomassie Plus Assay Kit, Thermo Fisher Scientific; Waltham, MA, USA). Samples were diluted at 1:100 in phosphate buffer, and the manufacturer’s instructions were followed. A plate spectrophotometer (Sparks 10M multimode microplate reader, Tecan Trading AG, Mannedorf, Switzerland) was used to analyze absorbance at 495 nm wavelength. 

Colostrum insulin was analyzed in duplicate samples using a porcine insulin ELISA kit (cat no. 10-1200-01; Mercodia AB; Winston Salem, NC, USA). Insulin was measured in both homogenate and skimmed colostrum samples. Intraplate variation was 4.75%. 

### 2.3. Neonate Plasma

#### 2.3.1. Protein

Plasma protein was measured in duplicate using the Bradford Assay (Bio-Rad Laboratories, Inc., Hercules, CA, USA) following manufacturer instructions. Prior to analysis, plasma was diluted 1:100 with phosphate-buffered saline. Intraplate CV was 3.65%.

#### 2.3.2. Insulin

Plasma insulin was analyzed in duplicate samples using a porcine insulin ELISA kit (cat no. 10-1200-01; Mercodia AB; Winston Salem, NC, USA), following manufacturer instructions. Intraplate variation was 4.75%.

#### 2.3.3. Glucose

Plasma glucose was determined using Autokit Glucose (Fujifilm Wako Diagnostics USA Corporation, Mountain View, CA, USA) following manufacturer instructions. Intraplate CV was 4.84%.

#### 2.3.4. Free Amino Acids

Free amino acid content of neonate plasma was analyzed using liquid chromatography-tandem mass spectrometry (LC/MS-MS) in Purdue University’s Bindley Biosciences Metabolite Profiling Facility. Briefly, 10 μL of amino-butyric acid at a concentration of 1 μg/uL and 25 μL of 100% trichloroacetic acid (TCA) solution were added to 100 µL of plasma. Samples were incubated for 10 min at 4 °C followed by centrifugation at 14,000× *g* for 10 min. The supernatant was collected and stored at −20 °C until analysis. Just prior to liquid chromatography, 100 µL of acetonitrile (ACN) was mixed with 100 µL of supernatant. Liquid chromatography was performed using Intrada Amino Acid 3 μm, 2 × 150 mm column (Imtrakt USA, Portland, OR, USA) connected to an Agilent 6470 QQQ LC-MS/MS system (Agilent, Santa Clara, CA, USA). Acetonitrile with 0.3% of formic acid and acetonitrile with 100 mM ammonium formate solution (20:80 *v*/*v*) were used as mobile phases.

### 2.4. Histological Analysis of Mammary Gland Development

All tissue preparations for histological analysis were done by the Purdue University Histology Research Laboratory. Mammary tissues were fixed in 10% neutral buffered formalin for 24 h and transferred to PBS until processing for paraffin embedding. Paraffin processing was done in a Sakura Tissue-Tek VIP6 tissue processor for dehydration through graded ethanols, clearing in xylene and infiltration with Leica Paraplast Plus paraffin. After processing, tissues were embedded in Leica Paraplast Plus paraffin. Tissue sections were taken at a thickness of 4 µm using a Thermo HM355S microtome. Sections were mounted on charged slides and dried for 30–60 min in a 60 °C oven. After drying, all slides were deparaffinized through 3 changes of xylene and rehydrated through graded ethanols to water in a Leica Autostainer XL. For hematoxylin and eosin (H&E) staining of tissues, the Leica Autostainer XL was used. Tissue sections were stained in Gill’s II hematoxylin, blued and counterstained in an eosin/phloxine B mixture. Finally, tissues were dehydrated, cleared in xylene and cover-slipped in a toluene-based mounting media (Leica MM24).

H&E-stained tissues were used to measure the proportion of epithelial tissue within the parenchymal compartment. First, ImagePro Plus 5.1 (Media Cybernetics) was used to capture histological images in conjunction with a Nikon Eclipse 50i microscope (Nikon Inc., New York, NY, USA; Evolution MP, Media Cybernetics Inc., Rockville, MD, USA). Multiple images of H&E stained tissue were captured at 10× magnification to encompass the entire parenchymal area of the gland for each animal. The parenchymal area was defined for this study as the epithelial cells of the terminal ductal lobular units (TDLU) and associated ducts along with intralobular and interlobular stroma. To create a panorama of the entire parenchymal area of the cross-section, images were merged into a single image using Adobe Photoshop (V 22.1.0, Adobe). ImageJ was used to measure the area in the tissue section (Figure 2). The “Draw/Merge: Trace” tool was used to first select parenchymal tissue and calculate the area, then to trace and calculate the entire epithelial area of TDLU (epithelium plus lumen) and finally to trace around the lumen and calculate that area. The ratio of epithelium within parenchyma was calculated by subtracting the lumen from the epithelial area of the TDLU and then dividing this by parenchyma area, and this was defined as parenchymal epithelial area (PEA).

Tissue sections were also immunostained with KI67 to mark proliferating populations of cells. After deparaffinization, antigen retrieval was done with a TRIS/EDTA pH 9.0 solution in a BioCare decloaking chamber (Pacheco, CA, USA) at a temperature of 95 °C for 20 min. Slides were cooled for 20 min at room temperature and transferred to TRIS buffer with Tween 20 detergent (TBST). The rest of the staining was carried out at room temperature using a BioCare Intellipath stainer. Slides were incubated with 3% hydrogen peroxide in water for 5 min. Slides were rinsed with TBST and incubated in 2.5% normal goat serum for 20 min. Excess reagent was blown off, and Ki67 primary antibody (Cell Marque, 275R-16, Rocklin, CA, USA) was applied at a dilution of 1:100 (0.364ug/mL) for 30 min. The negative control slide was stained with Rabbit IgG (Vector Labs, I-1000, Burlingame, CA, USA) at a concentration of 1:5000 (1 µg/mL) for 30 min. Slides were rinsed twice in TBST, and a goat anti-rabbit secondary antibody (Vector Labs, MP-7451) was applied for 30 min. Slides were rinsed twice in TBST, and Vector ImmPACT DAB (Vector Labs, SK-4105) was applied for 5 min. Slides were rinsed in water and transferred to a Leica Autostainer XL(Wetzlar, Germany)or hematoxylin counterstain, dehydration and cover-slipping. 

Five images per gilt were taken at 200× magnification. Sections of jejunum tissue were used as a positive control for the specificity of KI67 staining for proliferating populations of cells. To determine the proliferation index of mammary epithelial cells and proliferating intralobular stroma cells in parenchymal tissue, an ImageJ plugin called Cell Count by GNU General Public License was utilized. As above, the parenchymal area was defined for this study as the epithelial cells of the TDLU with ducts and associated intralobular and interlobular stroma. The proliferation index of epithelial and stromal cells within parenchymal tissue was determined. All epithelial cells were positively stained for KI67, the five sections were counted, and the epithelial cells without staining were counted. Similarly, intralobular and interlobular stromal cells that were immunostained for KI67 were counted, and all cells not stained were counted. The total number of each cell type was determined, and then the number of proliferating epithelial or stromal cells was divided by the total of each type to determine the percent of proliferating cells.

All research assistants that analyzed histomorphology were blinded to treatment and day and trained by one individual on the approach to conducting analyses. For each animal, three research assistants analyzed histomorphic features, and data across the three researchers were averaged for final counts. Tissue was available for all animals that survived to postnatal day 7 of COL10 (*n* = 7). However, the quality of tissue collected for one COL20 piglet was not representative of parenchyma, so only six animals in this treatment were used for histological analysis. To determine relative changes in the proliferating index of epithelial and intralobular stromal cells in the parenchyma and parenchymal epithelial area between birth and postnatal day 7, tissue from the baseline group (*n* = 6) of gilts was also analyzed. 

### 2.5. Mass Isotopomer Distribution Analysis (MIDA) of DNA and Protein Synthesis (f) and Fractional Synthetic Rate

#### 2.5.1. Metabolic Labelling with Deuterium Oxide (D_2_O)

In order to obtain good quality data for mass isotopomer distribution analysis for proteome studies, greater than 2.5% steady-state enrichment of heavy water-deuterium oxide (D_2_O) is recommended [14]. To achieve this, the doses of D_2_O used were adapted from Lam et al. [17]. Metabolic labeling of newly synthesized deoxyribose and amino acid molecules was begun immediately after birth by administering a deuterium oxide bolus to animals via intraperitoneal injection (IP) of 0.9% NaCl in D_2_O (Millipore Sigma, Burlington, MA, USA), at 20 mL/kg of BW. A second bolus was given to animals 4 h later, using the same dose and route of administration. At 24 h postnatal, and then daily until postnatal day 7, piglets were orally gavaged with 10 mL/kg of BW deuterium oxide at 0600 h to maintain enrichment. Six unlabeled animals were administered equivalent doses of saline. Tissue and blood of the unlabeled group were used to determine the baseline mass isotopomer distribution. At birth and postnatal days 1, 3, 5 and 7, blood samples were collected by jugular puncture and plasma was isolated for analysis of D_2_O enrichment.

#### 2.5.2. Determination of D_2_O Enrichment in Plasma Using Gas Chromatography-Tandem Mass Spectrometry (GC-MS/MS)

To determine the percent of body water that was D_2_O, the approach of base-catalyzed exchange of hydrogen (deuterium) between water and acetone was used [18], with modifications by Purdue University’s Bindley Biosciences Metabolite Profiling Facility. Briefly, piglet plasma (20 μL) was mixed with 2 μL of 10 N NaOH and 4 μL of 5% (*v*/*v*) acetone in acetonitrile. The sample and standard curve (ranged from 0.5 to 16% of D_2_O in H_2_O) mixtures were incubated at room temperature overnight, after which the acetone portion was extracted by the introduction of 500 μL of chloroform and 0.5 g of anhydrous sodium sulfate (Na_2_SO4). Samples were centrifuged at 14,000× *g* for 1 min to precipitate the Na_2_SO4. The chloroform solution was transferred to a vial, and acetone was measured using GC/MS in Purdue University’s Metabolomics Core. Gas chromatography was carried out using an Agilent Select FAME column (CP7419, Agilent, Santa Clara, CA, USA) attached to a TSQ 8000 triple quadrupole GC-MS/MS (Thermo Fisher Scientific, Waltham, MA). The intensity of acetone was measured at 58 and 59 *m*/*z* and was used to calculate D_2_O percentage in plasma.

#### 2.5.3. Isolation of DNA from Tissue and DNA Hydrolysis

The procedures for DNA isolation and hydrolysis described were modified from the approach described by others [19] in the following way. Approximately 25 mg of mammary tissue from the parenchymal area was used for DNA isolation. Tissue was ground using a tissue homogenizer in DNA extraction buffer from the gMAX Mini Genomic DNA Kit (IBI Scientific, Dubuque, IA, USA). DNA was eluted from the column using 100 µL of TE buffer (tris-HCl + EDTA), and the concentration was measured with a nanodrop system. DNA was diluted with hydrolysis buffer (20 mM Tris-HCl, 100 mM NaCl, 20 mM MgCl_2,_ pH 7.9) to 1 µg in a total volume of 50 µL, and 50 µL of hydrolysis enzyme cocktail [benzonase (E-1014); phosphodiesterase I (P-3243) and alkaline phosphatase (P-7923); MilliporeSigma, Burlington, MA, USA) was added. The hydrolysis reaction was carried at 37 °C for 6 h. Then, samples were dried overnight and stored at −20 °C until analysis.

#### 2.5.4. Liquid Chromatography-Tandem Mass Spectrometry (LC-MS/MS) Analysis of Adenosine Isotopomer Distribution

D_2_O labels the deoxyribose moiety of dNTPs in replicating DNA through the de novo nucleotide synthesis pathway. The isotopic enrichment of the purine deoxyribonucleoside adenosine is then determined by LC-MS/MS. Briefly, samples were reconstituted in 100 µL of 5% MeOH/95% 5 mM ammonium formate. Molecule separation was carried out with 5 mM ammonium fumarate and 100% methanol as mobile phases in a Waters Atlantis T3, 3 μm, 2.1 × 50 mm column (186003717, Waters Corp., Milford, MA, USA) connected to an Agilent 6470 QQQ LC-MS/MS system (Agilent, Santa Clara, CA, USA). Multiple reaction monitoring (MRM) of the ribose portion of adenosine (dA) was measured based on the parental and product ions 251 → 117 *m*/*z* (M0). Ion combinations for M+1 and M+2 were identified and measured based on the identifications of 252 → 118 *m*/*z* and 253 → 119 *m*/*z*, respectively.

#### 2.5.5. Protein Hydrolysis

Preparation of protein hydrolysate for measuring global protein synthesis was done as described [15] with some modifications. Briefly, approximately 25 mg of parenchymal mammary tissue were placed in a 5 mL amber glass vial (Fisherbrand, Thermo Fisher Scientific, Waltham, MA, USA), and 1 mL of 6 M HCl was added under the fume hood. Samples were homogenized using the Fisherbrand 150 handheld tissue homogenizer (Thermo Fisher Scientific, Waltham, MA). The probe of the homogenizer was washed with sterile water between samples. Caps were placed in vials and incubated at 120 °C in a forced air oven (Model 414004-576, VWR International, West Chester PA, USA) for 24 h. Following incubation, samples were transferred to a 1.5 mL tube and centrifuged at 14,000× *g* for 10 min. The supernatant was transferred to a 1.5 mL tube and dried in a savant SPD 2010 speedvac concentrator (Waltham, MA, USA) overnight. The dried samples were stored at −20 °C until amino acid extraction.

#### 2.5.6. Amino Acid Extraction LC/MS Analysis of Isotopomer Distribution of Alanine

Dried protein hydrolysates were reconstituted by adding 300 µL of PBS and vortexing the samples, and 100 µL was transferred to a new 1.5 mL tube. Twenty-five µL of TCA (trichloroacetic acid, saturated solution, 1000 mg of TCA + 700 µL H_2_O) was added and samples vortexed to mix. Samples were then centrifuged at 14,000× *g* for 10 min, and 50 µL were transferred to a new tube, being careful to avoid black precipitate. Then 50 µL of acetonitrile was added, and samples were mixed well by vortexing. One hundred µL of this extract was used for LC/MS analysis of alanine.

The method used to determine the isotopomers of alanine was developed by Purdue University’s Metabolite Profiling Facility, Bindley Bioscience Center, through modification of the methods used to measure amino acids. In this method, an Intrada Amino Acid column was used for the liquid chromatography (LC), followed by a quadrupole mass spectrometer (MS). Alanine is retained to ~11.5 min of the run, and the mass spectrometry returns a precursor ion of 90 *m*/*z* and a product ion of 44 *m*/*z*. The fragment of 44 *m*/*z* (with chemical formula C_2_H_6_N) contains four hydrogens that can potentially be replaced by deuterium during the synthesis process. The precursor (alanine, C_3_H_7_NO_2_) and product (C_2_H_6_N) will increase mass equally as deuterium is added to the molecule. For this method, the LC/MS machine and software is programmed to measure the intensity/area of the peaks of molecules with precursor → product ion pair of 90 *m*/*z* → 44 *m*/*z*; 91 *m*/*z* → 45 *m*/*z*; 92 *m*/*z* → 46 *m*/*z*; 93 *m*/*z* → 47 *m*/*z*; and 94 *m*/*z* → 48 *m*/*z*; in order to measure the intensity/area of isotopomer (M) with no heavy isotopes (M0), one (M+1), two (M+2), three (M+3) and four (M+4), respectively. Appendix A shows the distribution of alanine M0, M+1, etc., in a sample from an unlabeled animal (blank) and eight samples from D_2_O labeled animals, with corresponding LC/MS spectra of samples.

#### 2.5.7. Mass Isotopomer Distribution Analysis (MIDA) of Adenosine and Alanine for Calculation of the Fraction (f) of DNA and Protein Newly Synthesize and the Fractional Synthetic Rate (FSR) of DNA and Protein

To determine the percent of newly synthesized DNA and proteins, the percent of enrichment (*p*) was calculated as the mean percentage of D_2_O enrichment from postnatal day one to day seven in each piglet. LC-MS/MS analysis of adenosine and alanine provided the intensities for M0 to M+2 and M0 to M+4, respectively. The percentage of M0 (%M0) at birth and day seven (%M0_7_) was calculated as the intensity M0 isotopomer divided by the total of the intensities for all isotopomers measured for adenosine and alanine. The difference between %M0 and %M0t is defined as EM0t [14].
EM0t = %M0t − %M0(1)

EM0t can be defined as the modifications on %M0 after a specific time (t) of exposure to D_2_O and is, in part, determined by the synthetic ratio (balance between synthesis and degradation) of DNA, as tissue cells are in a constant turnover of division and death. Each cell has a different turnover based on its biological functions and properties. Moreover, the speed of synthesis of dA, or FSR, can be modified by physiological events or experimental treatments. The %M_0_* (max %M_0_ when 100% of the DNA are de novo synthesized) can be calculated based on %M_0_, *p* and *n.* Where *p* is the probability of being labeled, which is the percent enrichment of D2O, and *n* is the number of hydrogens in a molecule that can be labeled (‘replace’) hydrogen. The following formulas for DNA synthetic rate were adapted from [16,17]:%M_0_* = (1 − *p*)*^n^** %M_0_(2)

After we know the %M_0_*, we are able to elucidate EM_0_*, similar to EM_0t_.
EM_0_* = %M_0_* − %M_0_(3)

EM_0_* is the %M_0_ when 100% of the DNA is turned over.

Once EM_0t_ and EM_0_* are determined, fractional synthesis (*f*) is calculated.
*f* = EM_0t_/EM_0_*(4)
FSR = −ln (1 − *f*)/t(5)

### 2.6. Statistical Analysis

All statistical analyses were performed in SAS (version 9.4; Cary, NC, USA). Amino acids that were below the detectable limit were assigned a value that was one-fifth the highest value, which was below the lowest value. Procs Mixed model was used with treatment run as the class. Normality was checked by running the residuals in the model. All statistical models included treatment and birthweight. Birthweight was excluded from the model if not significant (*p* < 0.05). The least-square means were performed using the Tukey-Kramer method, with a pairwise comparison. *p* < 0.05 was considered significant, with *p*-value > 0.05, but ≤ 0.1 discussed as a tendency. Bodyweight and average daily gain were run with Procs Mixed with treatment and day as fixed effects with the day as a repeated measurement. Treatment by day interactions and birthweights were included in the model and removed if not significant (*p* < 0.05). Proc corr was used for correlations.

## 3. Results

### 3.1. Colostrum Composition and the Effect of Colostrum Intake on Neonate’s Plasma Insulin, Glucose, Protein and Amino Acid Concentrations

The homogenate colostrum sample that was fed to neonates was 10.1% fat and 9.8% protein, and the insulin concentration was 289 milli-international units per liter (mIU/L) in whole colostrum and 312 mIU/mL in skimmed colostrum. The birth weight of COL10 animals was not different from COL20 (Table 1). The COL20 animals gained significantly more weight over the 24 h colostrum feeding period than COL10 gilts (*p* = 0.03). From day two postnatal to study completion on postnatal day seven, there was no significant difference in average daily gain (ADG) between COL10 and COL20 animals. The final weight of COL20 was numerically greater than COL10 (Table 1).

Plasma insulin concentration was not different between COL10 and COL20 at 24 h after birth (1.84 vs. 1.65 mLU/L; *p* > 0.05), and glucose concentration at 24 h postnatal was not different between groups (Table 2).

There was no difference in total protein concentration between groups; however, the concentration of several amino acids was different between the treatments (Table 3). COL20 piglets had greater concentrations of all branched-chain amino acids (BCAA) (Ile, Leu, Val) than COL10 piglets at 24 h postnatal (*p* < 0.05; Table 2). COL20 piglets also had greater concentrations of the essential amino acids (EAA) Met and Phe (*p* < 0.05). While COL20 piglets had a trend to have greater Thr at 24 h of postnatal (*p* = 0.07), there was no difference between treatments for the EAA of Arg, His, Lys or Trp (*p* > 0.05). For non-essential amino acids, COL20 piglets had greater concentrations of Asp, Gln and Pro compared to COL10 piglets at 24 h postnatal (*p* < 0.05). There was no difference in Ala, Asn or Cys between the treatments 24 h postnatal.

### 3.2. Effect of Day and Colostrum Intake on Proliferation Index of Epithelial and Stromal Cells in Parenchyma and Proportion of Epithelial Tissue in Parenchymal Component of the Mammary Gland

There was no difference (*p* > 0.05) between COL10 and COL20 treatments in the proportion of epithelial tissue in the parenchymal compartment (PEA) of gilt mammary glands, nor was there a difference between baseline levels on day zero with the epithelial area on day seven. Both epithelial and stromal cells were immunostained for KI67 in the mammary parenchymal area, with KI67 positive stromal cells in both the intralobular and interlobular stroma (Figure 3). There was no difference in the proliferation index of epithelial (percent proliferating) or stroma cells in mammary parenchymal tissue between COL10 and COL20 piglets (*p* > 0.05; Figure 4). The epithelial cell proliferation index was greater in mammary tissue isolated from animals at birth versus the rate of proliferation in mammary tissue of COL10 and COL20 animals, which was isolated on postnatal day seven. Meanwhile, there was no significant difference in stromal cell proliferation rate nor PEA in the tissue between birth and day seven.

### 3.3. Effect of Colostrum on Amount (f) and Rate (FSR) of Mammary DNA and Protein Synthesis over the First Week Postnatal

Analysis of percent deuterium oxide in piglets’ plasma across the seven days of labeling indicated that the dosing regimen achieved the goal of greater than 2.5% D_2_O. The average enrichment (mean ± standard deviation) of COL10 animals from day one to seven postnatal was 3.48% ± 1.21, and COL20 averaged 3.38% ± 0.82 (Figure 5). The yield of DNA per unit of mammary parenchymal tissue, although numerically greater in COL20 (0.66 ± 0.08 ng DNA/mg tissue) treated gilts, was not different (*p* = 0.2) from COL10 (0.55 ± 0.21 ng DNA/mg tissue) animals (Table 4).

There were no differences between the fraction (f) of newly synthesized DNA over the seven days of labeling in mammary parenchyma between COL10 (29 ± 5%) and COL20 (29 ± 5%) piglets. The fractional synthetic rate (FSR) of DNA was approximately 5%, indicating approximately 5% of the cells were turning over each day in parenchymal tissue (*p* > 0.05). The f of the new protein in parenchyma over the seven days of labeling and the FSR of protein was not different between the treatments, with the f at 70% in COL10 and 68% in COL20 gilts and FSR at 18% for both groups (Table 4). The ratio of protein f to DNA f reflected the amount of newly synthesized protein synthesized per new DNA complement (i.e., new protein per new cell synthesized) over the seven days and was approximately 3:1 for both treatments. The ratio of protein FSR to DNA FSR was also not different between treatments and was 4.36 ± 0.99 for COL10 and 4.47 ± 0.99 for COL20.

The relationship between mammary morphological features and mammary DNA and protein f and FSR were investigated (Table 5). A relatively strong relationship (*r* = 0.86; *p* < 0.0001) between percent KI67 labeled stroma, and epithelial cells was evident on postnatal day seven. PEA was related to percent KI67 labeled stroma (*r* = 0.75; *p* < 0.001) and KI67 labeled epithelial cells (*r* = 0.66; *p* = 0.01). The percent of KI67 labeled epithelial cells on postnatal day seven was positively (*p* < 0.05) associated with protein f (*r* = 0.61) and FSR (*r* = 0.63). There was a trend for a relationship between parenchymal epithelial area and the fraction of newly synthesized DNA (*r* = 0.48; *p* = 0.09), as well as a trend for inverse relationships between parenchymal epithelial area and the ratio of protein to DNA f (*r* = −0.49; *p* = 0.09) and FSR (*r* = −0.48; *p* = 0.1). Consistent with the relationship between f and FSR, these showed strong (*p* < 0.001) correlations with each other within DNA and protein. The positive relationships between protein and DNA f and FSR were also strong (*p* < 0.001). There were also strong negative relationships between DNA f and FSR with the ratio of protein/DNA f and FSR. The relationship between protein f and FSR and this ratio was also negative.

Correlation analysis was run to determine if growth and morphology variables of all animals were related to markers of perinatal nutritional environment such as birthweight, growth variables (i.e., average daily gain, crown-rump length), plasma glucose, insulin, protein and amino acids at 24 h after birth. No relationship was found between birthweight and the f or FSR of DNA and protein. There was also no relationship between birthweight and histomorphic features of gilt mammary glands on postnatal day seven. However, average daily gain was correlated with PEA (*r* = 0.54; *p* = 0.05) and f and FSR of both DNA and protein (*p* < 0.01; Table 4). There was also a significant relationship between DNA f (*r* = 0.60; *p* = 0.02) and FSR (*r* = 0.61; *p* = 0.02) and crown-rump length on day seven (Table 5). Ongoing analysis indicated growth of other tissues, including *longissimus dorsi* muscle from the same animals that mirrored that of the mammary parenchyma.

There was no relationship between any mammary variables and plasma levels of glucose, insulin or protein at 24 h postnatal. The relationship between individual plasma amino acid levels and mammary variables was found only for plasma lysine and glutamine, and so only these amino acids are listed in Table 5. Plasma lysine level at 24 h postnatal was positively related to mammary DNA f and FSR (*r* = 0.57; *p* = 0.03 and *r* = 0.57; *p* = 0.03, respectively, Table 5). There was an inverse relationship between lysine levels and the ratio of protein to DNA f (*r* = −0.56; *p* = 0.04) and FSR (*r* = −0.56; *p* = 0.04) and lysine levels. Plasma lysine levels were also positively correlated with average daily gain across the seven days (*r* = 0.54, *p* = 0.05). Plasma glutamate levels were negatively associated with the parenchymal epithelial area (PEA; *r* = −0.55, *p* = 0.05), and there was a tendency (*p* ≤ 0.1) for a positive relationship between plasma glutamate and the ratio of protein to DNA f (*r* = 0.47) and FSR (*r* = 0.48).

## 4. Discussion

The data collected supports the relationship between factors indicative of perinatal nutritional environment and mammary growth and development over the first week postnatal. In particular, plasma lysine level at 24 h postnatal was positively related to average daily gain, the fraction of newly synthesized DNA (f) in mammary parenchymal tissue over the first seven days postnatal, and the fractional synthetic rate of DNA in mammary parenchyma. Plasma lysine was also inversely related to the ratio of protein to DNA f and FSR. This relationship, as posited in the introduction, may reflect that higher lysine levels favored a greater degree of cell division versus cells leaving the cell cycle and differentiating. The relationships between nutritional environment and mammary development were found despite the fact that colostrum dose was not related to any of the variables used to evaluate mammary development. There may not be an effect of colostrum dose on variables measured. In light of this possibility, it is interesting to note that the amount of DNA isolated per unit of mammary parenchymal tissue was numerically higher in COL20 versus COL10 animals. This finding suggests that the level of colostrum intake may affect the number of cells in parenchyma. Analysis of DNA content at an earlier time point is needed to determine this. Moreover, future studies using tools like single-cell RNA-seq would help in understanding whether the amount of colostrum consumed affects the developmental program of subpopulations of cells within the gland. The lack of an effect may also have been related to the study design. Different doses of colostrum resulted in COL20 animals weighing significantly more after the 24 h of colostrum feeding, and these differences were maintained to postnatal day seven [13]. However, returning piglets to birth litters likely had unmeasured impacts on perinatal nutrition. One piglet in each group died by crushing, and the growth rates were highly variable after return to litters. Bottle feeding and returning neonates to litters where they competed for access to milk, likely differentially stressed animals and contributed to piglets’ nutritional environment. Moreover, sow milk quality likely varied across litters. With this in mind, circulating levels of nutrients and gross measures of growth were used as corollaries of the postnatal nutritional environment.

Variables used to assess mammary development over the first week postnatal correlated with each other, supporting the potential for underlying relationships. PEA reflects the relative amount of epithelial tissue in the gland and was positively correlated with the percent of epithelial and stromal cells proliferating on postnatal day seven, as well as the average daily gain. Percent of proliferating epithelial cells was also related to protein synthesis over the first week postnatal. The different degrees of mammary development between animals in our study were most likely attributed to the postnatal environment and not prenatal factors, such as mammary histomorphology and DNA and protein f and FSR variables were not correlated with gilt birthweight. Whereas PEA and f and FSR of mammary DNA and protein were correlated with average daily gain. Moreover, crown-rump length on postnatal day seven was correlated with the fraction of newly synthesized DNA and DNA FSR. Although it is important to note that the nutritional environment in late gestation carries over to postnatal growth performance, as discussed below, in regard to lysine. Since gilts were matched by birthweight when assigned to treatments, the relationship to these variables would be minimally expected.

The relationship between average daily gain and mammary growth metrics supports mammary growth as isometric to body growth in the first postnatal week. The relationship between average daily gain to the mammary parenchymal epithelial area is particularly intriguing with regard to our hypothesis. This relationship likely indicates that the more adequately nourished the gilt is, the more resources are partitioned to growing the secretory component of the gland. A higher proliferative index of epithelial cells was found in mammary tissue isolated at birth compared to tissue from COL10 and COL20 animals on postnatal day seven. The proliferative population at birth may be particularly sensitive to nutritional growth cues in the gilt’s postnatal environment. Studies of mature pigs support that the nutritional environment affects mammary development [20,21].

As an essential amino acid, lysine availability is limiting to porcine growth. Studies of the effect of lysine availability on suckling neonates through maternal milk found diets with a 20% deficiency in lysine content had reduced litter growth by 8–10% [20]. At the same time, litter growth increased by 2.35 times with a 2.90-fold increase in amino acids in the sow’s diet. Improved growth was directly related to the increased intake of lysine and other amino acids by neonates [20]. Thus, our finding that plasma lysine levels at 24 h were related to the average daily gain of neonates is consistent with its availability to neonates limiting growth. Currently, it is not known what led to the varying concentrations of circulating lysine across gilts, as levels of lysine were not related to the 24 h dose of colostrum. Increasing lysine content of sow diet in late gestation increased the total number born alive and birth weight of piglets [22]. Similarly, increasing the lysine and fat content of sow diets in late gestation diets improved the overall performance of litters [23]. It also increased sow colostrum production. Although all sows in the present study were on the same diet, the efficiency of absorption of lysine by sows and placental lysine transfer to gilts during their late fetal growth may be a factor. Further research in this area is needed, as lysine levels at 24 h postnatal were also positively related to the fraction of newly synthesized DNA and the fractional synthetic rate of DNA in mammary parenchymal tissue over the first seven days postnatal. At the level of the cell, nutrients, including amino acids, regulate gene expression [24]. The mTORC1 signaling pathway in cells functions to integrate nutrient availability, growth factor signaling and developmental cues to regulate growth [25]. The production of proteins, lipids and nucleotides need to increase for cells to grow and divide, while catabolic pathways such as autophagy need to be suppressed. mTORC1 regulates all of these processes [25]. Amino acids and positive cellular energy status activate mTORC1, whereas lack of amino acids or energy inhibits its activity [26]. Thus, the association of lysine with the total amount of DNA synthesized and DNA synthetic fractional synthetic rate likely reflects that it is indicative of nutrient-amino acid availability that can be used for cellular growth, and in particular, nucleotide synthesis. 

The body does not store amino acids, so muscle protein is mobilized to produce free amino acids [24]. Protein undernutrition decreases the plasma level of most essential amino acids and causes adjustments in physiological functions, with a primary consequence of feeding a low protein diet resulting in the inhibition of growth [24]. The lower levels of essential amino acids and lower growth of COL10 gilts relative to COL20 animals likely reflect adaptations of COL10 animals to the undernutrition they experienced over the first 24 h of the experiment. Furthermore, lower in COL10 animals, there were several non-essential amino acids, including glutamine. For maximal growth performance, pigs require dietary glutamine [27]. Studies of neonatal pigs found that nearly all glutamate and glutamine feed was metabolized by the gut, so that glutamate and glutamine in the body must derive almost entirely from synthesis de novo [28]. Glutamine and glutamate are precursors and products of each other, with reactions catalyzed by glutamine synthetase and glutaminase. Circulating glutamate levels appear to be resistant to large variations, with levels relatively constant in experimental manipulations of diet and energy [29]. Part of this persistence may be due to the sensitivity of hepatic glutaminase to metabolic state and its transcriptional stimulation induced by starvation. Meanwhile, there is little effect of metabolic state on glutamine synthetase [29]. This may explain the significantly lower levels of glutamine in COL10 versus COL20 animals. Moreover, although seemingly counterintuitive, the negative relationship between glutamate and PEA may reflect that lower energy in the postnatal environment negatively impacts mammary epithelial expansion.

In the present study, the effect of returning piglets to the birth litter after 24 h of bottle feeding was not controlled, other than by matching treatments by litter. Differences in milk composition between sows and competition between piglets could influence developmental trajectory in the mammary tissue. In addition, this study did not consider other bioactive factors in colostrum, like fatty acids or hormones. Future studies aiming to understand the programming effects of colostrum need to control for maternal effects on piglet development after colostrum feeding by returning piglets used in the study to a common sow. This approach would limit competition of study piglets with established piglets that have experience nursing from the dam. Moreover, to control for nourishment versus bioactive factors in milk, future studies should use a nutrient-matched formula that represents the energy provided by colostrum but does not contain bioactive components. Finally, future studies should include more time points for sample and tissue collection to increase the understanding of the mechanisms involved in neonatal programing. 

## 5. Conclusions

Overall, this study found plasma lysine levels at 24 h postnatal were positively related to average daily gain and DNA synthesis in mammary parenchyma over the first week postnatal. This relationship supports that the nutritional environment affects early mammary development. Moreover, data support the potential that higher lysine levels in the perinatal period favored a greater degree of cell division versus differentiation in mammary parenchyma of neonatal pigs. Further investigations are needed to determine if manipulating the level of lysine provided to neonatal pigs affects postnatal mammary development and distribution of cell types in the parenchyma. 

## Figures and Tables

**Figure 1 animals-11-02867-f001:**
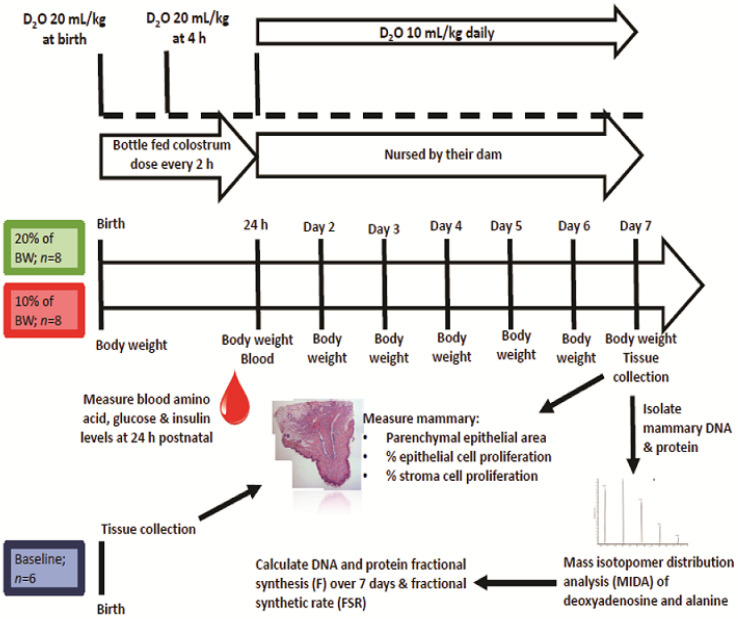
Schematic of study design.

**Figure 2 animals-11-02867-f002:**
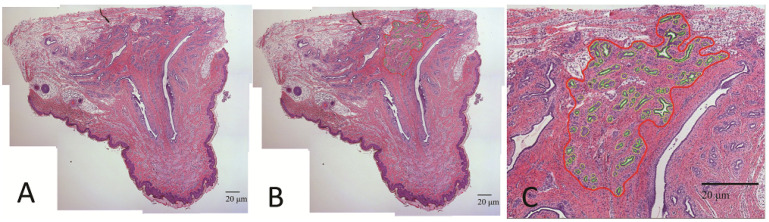
Histological section of teat and mammary tissue of 7-day postnatal gilt. (**A**) Teat and mammary tissue were excised from 7-day postnatal gilts, and images were captured at 200×. (**B**,**C**) Illustrate the selection of the mammary parenchymal area (red outline) and mammary epithelium (green outline) within this region for calculation of parenchymal epithelial area (PEA).

**Figure 3 animals-11-02867-f003:**
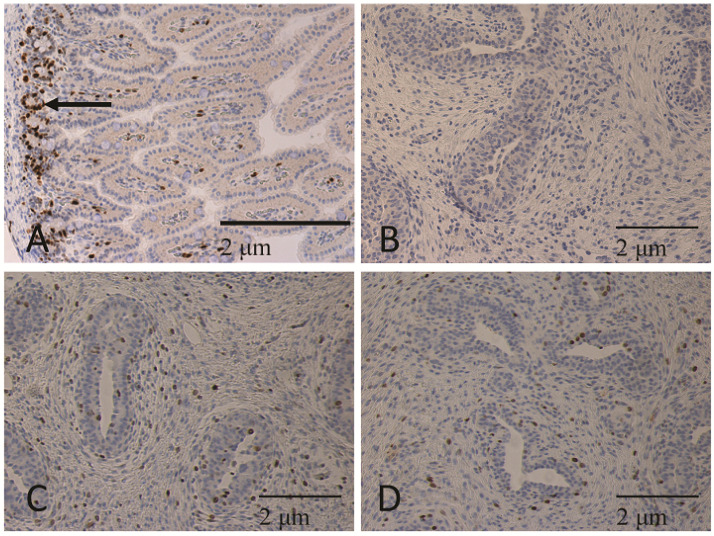
Tissues immunostained for KI67 (brown) and counterstained with hematoxylin. (**A**) Jejunum was collected from gilts and used as a positive control for KI67 labeling of proliferating cell populations in tissues. Arrow indicates crypt region of villi. Immunostaining of KI67 shows that epithelial cells within the crypt region of the villi were immunostained, whereas those lining the villi of the jejunum were mostly lacking positive cells. (**B**) Negative control mammary tissue was incubated with rabbit IgG rather than primary antibody and show a lack of brown nuclei. Mammary tissue from (**C**) COL10 and (**D**) COL20 gilts immunostained with KI67 have positively stained cells in the epithelial and stromal components of the parenchymal compartment. Images were captured at 200×.

**Figure 4 animals-11-02867-f004:**
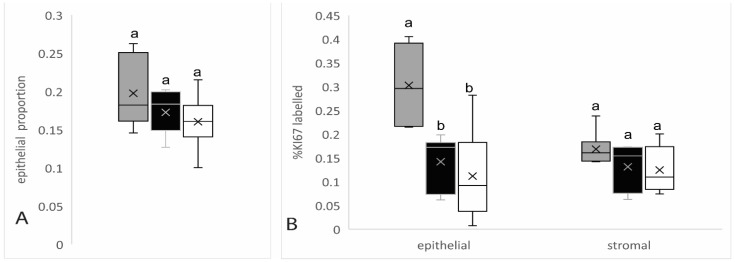
The effect of day and colostrum treatment on the proportion of parenchyma composed of epithelial tissue and percent epithelial and stromal cell proliferation index. (**A**) Proportion of parenchymal tissue composed of epithelium (parenchymal epithelial area; PEA) and (**B**) percent of epithelial and stromal cells in parenchymal compartment stained with KI67 as a marker of proliferating population in mammary tissue of baseline (*n* = 6; gray) animals collected immediately after birth and in neonatal piglets fed 10% (COL10; *n* = 7; black) or 20% (COL20; *n* = 6; white) of birth weight of colostrum over the first 24 h postnatal, returned to sow to suckle and then euthanized on day seven postnatal. Within the boxplot, the ‘x’ indicates mean, and line indicates median. Different letters indicate a significant difference at *p* < 0.05.

**Figure 5 animals-11-02867-f005:**
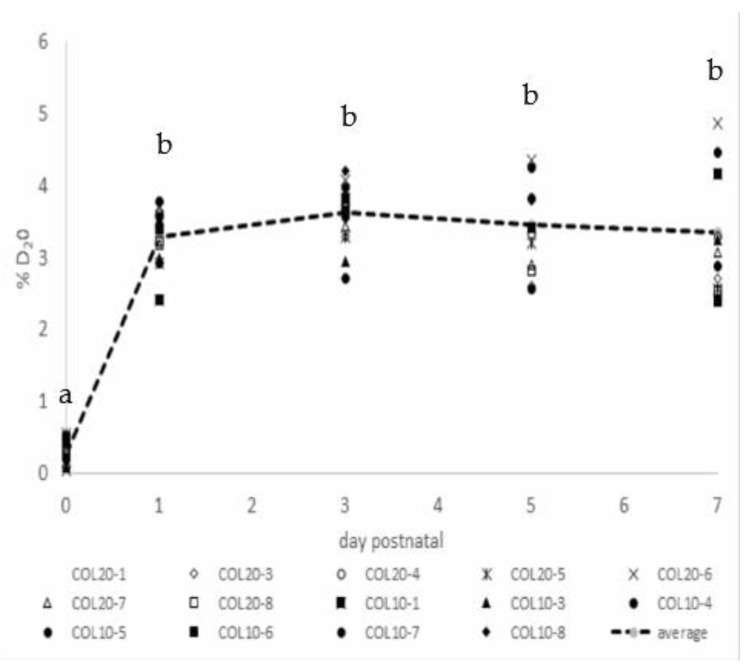
Percent deuterium oxide (D_2_O) in plasma of neonatal gilts at birth (day 0) and 1, 3, 5 and 7 days postnatal. Open and cross-hatched symbols represent percent deuterium in COL20, and filled symbols represent COL10 animals across the five days sampled. The dashed line and gray markers represent the mean percent deuterium across all animals. Different letters indicate a significant difference at *p* < 0.05.

**Table 1 animals-11-02867-t001:** Birth body weight and the effect of colostrum intake at a rate of 10% (COL10) and 20% (COL20) over the first 24 h after birth on weight (day 1) and average daily gain (ADG; birth to day 1), and weight and average daily gain one day following the return to birth litter (day 2 and day 1–2), weight and growth to study termination (day 7; day 2 to 7 and birth to day 7) and postnatal day 7 crown rump length.

		COL10		COL20		
Variable	Day Postnatal	Mean	Std Error	Mean	Std Error	*p*
weight (Kg)	birth	1.48	0.08	1.51	0.07	ns
	day 1	1.49	0.06	1.65	0.08	ns
	day 2	1.52	0.07	1.66	0.10	ns
	day 7	2.40	0.21	2.54	0.27	ns
Crown-rump (cm)	day 7	28.43	2.69	29.29	2.91	ns
ADG (Kg)	birth-day 1	0.01	0.02	0.14	0.02	0.03
	day 1 to 2	0.06	0.02	0.01	0.03	ns
	day 2 to 7	0.16	0.01	0.15	0.02	ns
	birth to day 7	0.14	0.02	0.14	0.03	ns

**Table 2 animals-11-02867-t002:** Insulin, glucose and protein plasma concentration in neonatal piglets fed 20% (COL20, *n* = 8) or 10% (COL10, *n* = 8) of birth weight of colostrum over the first 24 h after birth.

Hormone/Nutrient-	COL10	COL20	SEM	*p*
Insulin (mUI/L)	1.84	1.65	0.90	0.83
Glucose (mg/dL)	75.25	81.45	7.89	0.21
Protein (mg/mL)	31.68	41.04	5.75	0.13

**Table 3 animals-11-02867-t003:** Amino acid plasma concentration (ng/mL) in neonatal piglets fed 10% (COL10, *n* = 8) or 20% (COL20, *n* = 8) of birth weight of colostrum over the first 24 h after birth.

Amino Acid	COL10	COL20	SEM	*p*
Ala	288.8	285.2	17.5	0.87
Arg	84.1	81.6	10.9	0.80
Asn	110.8	110.1	10.4	0.90
Asp	10.4	15.9	1.7	0.04
Cys	17.7	17.8	0.2	0.56
Gln	698.6	1081.9	113.5	<0.01
Glu	83.3	106.7	11.2	0.11
Gly	319.4	239.7	20.5	<0.01
His	873.5	687.5	111.7	0.75
Ile	215.7	294.3	26.1	0.01
Leu	39.3	72.7	13.1	0.03
Lys	1555.9	1696.9	328.8	0.46
Met	33.8	79.0	9.9	<0.01
Phe	173.7	211.9	10.0	0.02
Pro	486.1	776.3	49.7	<0.01
Ser	201.1	175.0	10.7	0.11
Thr	133.4	202.6	29.9	0.07
Trp	59.9	66.6	6.7	0.13
Tyr	458.1	549.1	35.9	0.07
Val	444.6	526.4	32.1	0.05

**Table 4 animals-11-02867-t004:** DNA yield, mean fraction (f) and fractional synthetic rate (FSR) of DNA and protein and the ratio of these factors in mammary parenchymal tissue over the first seven days postnatal in gilts fed 10% (COL10, *n* = 7) or 20% (COL20, *n* = 7) of birth weight colostrum over the first 24 h after birth.

Variable	COL10	COL20	SEM	*p*
DNA content (ng/mg tissue)	0.55	0.66	0.15	0.20
Mammary Protein f (%)	0.70	0.68	0.06	0.81
Mammary Protein FSR (%/day)	0.18	0.18	0.02	0.96
Mammary DNA f (%)	0.29	0.29	0.05	0.98
Mammary DNA FSR (%/day)	0.05	0.05	0.01	0.95
Mammary Protein f: DNA f	2.97	3.13	0.78	0.87
Mammary Protein FSR: DNA FSR	4.36	4.47	0.99	0.94

**Table 5 animals-11-02867-t005:** Correlation analysis of the relationship between corollaries of perinatal nutrition (plasma concentration of insulin, protein, glucose, lysine and glutamate at 24 h postnatal; birth weight, crown rump length on postnatal day seven, and average daily gain between birth and day seven) and variables used to evaluate mammary growth.

Variable	r/p	Histological Features			DNA		Protein		Protein/DNA	
		PEA	% KI67		f	FSR	f	FSR	f	FSR
			Epi	Stroma						
PEA	*r*		0.66	0.75	0.48	0.46	0.41	0.32	−0.49	−0.48
	*p*		0.01	0.00	0.09	0.11	0.16	0.29	0.09	0.10
% KI67-epi	*r*	0.66		0.86	0.38	0.37	0.61	0.63	−0.31	−0.22
	*p*	0.01		0.00	0.19	0.22	0.03	0.02	0.31	0.48
% KI-stroma	*r*	0.75	0.86		0.28	0.25	0.36	0.36	−0.36	−0.32
	*p*	<0.0001	<0.0001		0.35	0.40	0.22	0.22	0.23	0.29
DNA f	*r*	0.48	0.38	0.28		1.00	0.87	0.82	−0.90	−0.86
	*p*	0.09	0.19	0.35		<0.0001	<0.0001	0.0003	<0.0001	<0.0001
DNA FSR	*r*	0.46	0.37	0.25	1.00		0.86	0.82	−0.88	−0.84
	*p*	0.11	0.22	0.40	<0.0001		<0.0001	0.0003	<0.0001	0.0002
protein f	*r*	0.41	0.61	0.36	0.87	0.86		0.98	−0.70	−0.59
	*p*	0.16	0.03	0.22	<0.0001	<0.0001		<0.0001	0.01	0.03
protein FSR	*r*	0.32	0.63	0.36	0.82	0.82	0.98443		−0.65	−0.54
	*p*	0.29	0.02	0.22	0.0003	0.0003	<0.0001		0.01	0.05
protein/DNA f	*r*	−0.49	−0.31	−0.36	−0.90	−0.88	−0.70	−0.65		0.99
	*p*	0.09	0.31	0.23	<0.0001	< 0.0001	0.006	0.012		<0.0001
protein/DNA FSR	*r*	−0.48	−0.22	−0.32	−0.86	−0.84	−0.59	−0.54	0.99	
	*p*	0.10	0.48	0.29	<0.0001	0.0002	0.03	0.05	<0.0001	
birthweight	*r*	0.33	−0.14	−0.02	0.00	0.01	−0.31	−0.34	0.07	−0.02
	*p*	0.27	0.65	0.94	1.00	0.97	0.29	0.24	0.80	0.94
crown rump day 7	*r*	0.44	0.04	0.04	0.60	0.61	0.37	0.31	−0.41	−0.44
	*p*	0.14	0.89	0.89	0.02	0.02	0.20	0.28	0.14	0.11
insulin	*r*	−0.08	0.13	0.17	0.14	0.13	0.10	0.12	−0.19	−0.19
	*p*	0.80	0.67	0.58	0.64	0.65	0.73	0.68	0.52	0.51
average daily gain	*r*	0.54	0.23	0.17	0.87	0.88	0.68	0.60	−0.68	−0.67
	*p*	0.05	0.44	0.58	<0.0001	<0.0001	0.006	0.02	0.009	0.008
glucose	*r*	−0.32	0.04	−0.10	0.24	0.23	0.28	0.37	−0.27	−0.24
	*p*	0.29	0.90	0.74	0.41	0.43	0.33	0.20	0.36	0.42
protein	*r*	0.42	0.43	0.27	0.25	0.26	0.38	0.36	0.05	0.10
	*p*	0.15	0.14	0.37	0.39	0.37	0.18	0.21	0.86	0.73
lysine	*r*	0.41	0.27	0.45	0.57	0.57	0.42	0.37	−0.56	−0.56
	*p*	0.16	0.38	0.12	0.03	0.03	0.13	0.19	0.04	0.04
glutamate	*r*	−0.55	−0.39	−0.33	−0.37	−0.35	−0.28	−0.25	0.47	0.48
	*p*	0.05	0.18	0.27	0.19	0.21	0.33	0.39	0.09	0.09

## Data Availability

All data will be made available upon request to the corresponding author.

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
