# Peer review of "Mammary Development in Gilts at One Week Postnatal Is Related to Plasma Lysine Concentration at 24 h after Birth, but Not Colostrum Dose"

_animals, 2021, doi:10.3390/ani11102867_

Round 1

Reviewer 1 Report

The study appears to bring new, interesting data concerning the role of nutrition (different % of colostrum) during the first 24h after birth, taking into consideration of amino acid concentration in the plasma of porcine neonates. Overall, the results are interesting and, in my opinion, valuable, nevertheless, after careful insight I have some comments and doubts.

Lines 85-88 - Authors mentioned that the ratio of protein to DNA synthesis is used as an indicator of stem-progenitor cells. Without the characterization of mammary epithelial stem/progenitor cells markers ( i.e. K8, K18, Muc1, K5, K14, SMA, CD10)  this parameter can only describe the relative amount of cell division.

In the introduction, there is a lack of information on why the Authors focus on lysine not on the other aa (lack of supportive references), which is highlighted in the title of the manuscript. The title of the manuscript should be changed, and must include also the information about the nutrition – “Plasma lysine concentration at 24 h after birth relates to mammary development in gilts at one week postnatal – study on females fed a high versus low dose of colostrum” or “Plasma lysine concentration at 24 h after birth relates to mammary development in gilts at one week postnatal – study on females fed different doses of colostrum”

Insulin Elisa kit, the cat. no. should be provided. The specificity of the ELISA kit, range of standard curve, etc. should be indicated. Was it specific for porcine samples? Did the authors perform the validation of the ELISA method?

Lines 190-192- Why did the Authors mention the fat and skim milk in the section regarding the measurement of the amino acid level in plasma?

The authors should characterize the features of  Ki67antibody. Was it specific for porcine cells-  according to information of Fallini et al. 1989 (Evolutionary conservation in various mammalian species of the human proliferation-associated epitope recognized by the Ki-67 monoclonal antibody), the Ki-67 did not stain proliferating cells in porcine tissues?

Lines228 and  231 - -it should be - Ki67 antibody and secondary antibody

The authors showed the Figures from the H&E staining. I would recommend presenting representative images for both 10%COL and 20%COL animals.

Lines 236-237 authors divided the analysis for epithelial and stromal cells, however in Fig. 1, and in the description of the methods, there is no explanation of how the Authors recognized the type of cells during the analysis of proliferation. In Lines 217-220, they described only epithelial area and parenchyma area, not cells!!
Lines 269- Lines 270, what does it mean COL15. Was it another control group? There is no explanation for COL15.
DNA hydrolysis (line 290), protein analysis(line315) – mammary tissue and parenchyma tissue, respectively. It should be precise which part of mammary tissues was used for DNA and protein isolation.
Authors could provide the exemplify of MS/LC spectra of alanine precursor and products as supplementary data.
Because the authors presented a proliferation index based on measurement Ki67 staining, the representative images (with negative controls) for all presented groups must be incorporated as another Figure into the manuscript.

In the statistics section, the Authors provided the information about the tendency  (P≤0.1), whereas in  Table 3  for Ser and Glu P=0.11. Therefore, the sentence ”There was also a trend for COL20 piglets to have a greater concentration of Glu, Thr, and Ser (P =  0.10).” is not true. Moreover, the Authors provided there the wrong information regarding Thr (p=0.07). This sentence should be removed from the manuscript or change.

Lines 510-512 – Sentence “Although not significant, in terms of discussion purposes, the negative relationships between plasma glutamate levels at 24 h postnatal and birthweight (r = -0.24), insulin (r = -0.17) glucose (r = - 0.14) are being reported.” – I couldn’t find these results in table 5, nor anywhere in the manuscript and this sentence should be removed.

 Fig.4 description. COL20-1 – there is no symbol.

 I am curious why authors focus on lysine, due to there were no differences between COL10 and COL 20%. Why Authors did not focus also on others aa. Did they perform the correlation analysis (table5) for others aa?

 Table 5 and lines 470-476 – there is no explanation for which animal (10, 20%COL or both) the correlation analysis was performed.

 Table 5 - why the Authors did not also perform the analysis of the relationship for the parenchymal area (indicated in Fig. 1).

 Table 5, description of the results, Discussion - According to my knowledge glutamate is not the same as glutamine – please check the text, or provide the information about the glutamate-glutamine cycle

 Discussion lines 521 - why the Authors stated that higher lysine levels favored a greater degree of a stem-progenitor cell? In the results section, there are no differences in lysine concentration in plasma of animals fed 10 and 20% COL. As was mentioned above, it cannot be determined as the degree of stem-progenitor cells.

Author Response

Response to review of: Manuscript ID: animals-1306258

Title: Plasma lysine concentration at 24 h after birth relates to mammary development in gilts at one week postnatal.

Dear Reviewers,

Thank you for taking the time to read and critique our article. Responding to your concerns improved the quality of the manuscript.

Please find below point-by-point response to your comments

Best, Theresa

Comments and Suggestions for Authors

Reviewer 1

  1. Lines 85-88 - Authors mentioned that the ratio of protein to DNA synthesis is used as an indicator of stem-progenitor cells. Without the characterization of mammary epithelial stem/progenitor cells markers ( i.e. K8, K18, Muc1, K5, K14, SMA, CD10)  this parameter can only describe the relative amount of cell division.

AU: We agree with the reviewer’s comment that the approach does not reflect stem/progenitor cells as they lack markers that specify these cells types. In the discussion section of the article we removed any reference to stem/progenitor cells.  We also removed argument that ratio was a marker of stem/progenitor abundance, as this would require tracer studies in conjunction with what was done.  We did leave content regarding progressive differentiation of cells is represented by more relative protein synthesis, versus less in less differentiated cycling cells (stem-progenitor) in the introduction.  

  1. In the introduction, there is a lack of information on why the Authors focus on lysine not on the other aa (lack of supportive references), which is highlighted in the title of the manuscript. The title of the manuscript should be changed, and must include also the information about the nutrition – “Plasma lysine concentration at 24 h after birth relates to mammary development in gilts at one week postnatal – study on females fed a high versus low dose of colostrum” or “Plasma lysine concentration at 24 h after birth relates to mammary development in gilts at one week postnatal – study on females fed different doses of colostrum”

AU: We did not focus on specific AA in the introduction, as this was a ‘discovery’ made as we analyzed data.  WE better address Lysine and glutamine/glutamate in the discussion. Thank you for suggesting titles, and pointing out the title of article was not comprehensive.  We changed it to: Mammary development in gilts at one week postnatal is related to plasma lysine concentration at 24 h after birth, but not colostrum dose.

  1. Insulin Elisa kit, the cat. no. should be provided. The specificity of the ELISA kit, range of standard curve, etc. should be indicated. Was it specific for porcine samples? Did the authors perform the validation of the ELISA method?

AU: the cat. no. was added, and the specificity of kit for porcine insulin was also indicated.

  1. Lines 190-192- Why did the Authors mention the fat and skim milk in the section regarding the measurement of the amino acid level in plasma?

AU: Thank you for catching this error; this was deleted from the manuscript.

  1. The authors should characterize the features of  Ki67antibody. Was it specific for porcine cells-  according to information of Fallini et al. 1989 (Evolutionary conservation in various mammalian species of the human proliferation-associated epitope recognized by the Ki-67 monoclonal antibody), the Ki-67 did not stain proliferating cells in porcine tissues?

AU: Sections of the jejunum of gilts were used as positive control for confirming specificity of KI67 staining of proliferating cell populations. Analysis demonstrated that the crypt cells were stained, whereas epithelial cells lining the villi were not.  There were also positively stained cells in the stromal component of the villi, which is what would be expected.  This information was added to the manuscript.  Image of KI67 staining of jejunum was added to the requested Figure of KI67.

  1. Lines228 and  231 - -it should be - Ki67 antibody and secondary antibody

AU: ‘antibody’ was added.

  1. The authors showed the Figures from the H&E staining. I would recommend presenting representative images for both 10%COL and 20%COL animals.

AU: We prefer not to add another set of images.  Figure 2 illustrates method of quantifying parenchymal epithelial area.  Since there was no difference between the colostrum doses, having representative images will not show any differences and so information would be redundant, and take up space.

  1. Lines 236-237 authors divided the analysis for epithelial and stromal cells, however in Fig. 1, and in the description of the methods, there is no explanation of how the Authors recognized the type of cells during the analysis of proliferation. In Lines 217-220, they described only epithelial area and parenchyma area, not cells!!

AU: Thank you for noting this.  The details were added to the text of the M&M and the legend of figure 2.

  1. Lines 269- Lines 270, what does it mean COL15. Was it another control group? There is no explanation for COL15.

AU: Thank you for noting this.  In order to calculate MIDA, a group of unlabeled animals (not exposed to D2O) are needed; for simplification we changed the name of the group to unlabeled, but missed this edit. We decided that COL15 would confuse reader and was a hold-over designation from the larger experiment.

  1. DNA hydrolysis (line 290), protein analysis(line315) – mammary tissue and parenchyma tissue, respectively. It should be precise which part of mammary tissues was used for DNA and protein isolation.

AU: edited as suggested.

  1. Authors could provide the exemplify of MS/LC spectra of alanine precursor and products as supplementary data.

AU: Added Supplementary Information S1.

  1. Because the authors presented a proliferation index based on measurement Ki67 staining, the representative images (with negative controls) for all presented groups must be incorporated as another Figure into the manuscript.

AU: Added figure, now designated Figure 3.

  1. In the statistics section, the Authors provided the information about the tendency  (P≤0.1), whereas in  Table 3  for Ser and Glu P=0.11. Therefore, the sentence ”There was also a trend for COL20 piglets to have a greater concentration of Glu, Thr, and Ser (P =  0.10).” is not true. Moreover, the Authors provided there the wrong information regarding Thr (p=0.07). This sentence should be removed from the manuscript or change.

AU: deleted sentence.

  1. Lines 510-512 – Sentence “Although not significant, in terms of discussion purposes, the negative relationships between plasma glutamate levels at 24 h postnatal and birthweight (r = -0.24), insulin (r = -0.17) glucose (r = - 0.14) are being reported.” – I couldn’t find these results in table 5, nor anywhere in the manuscript and this sentence should be removed.

AU: deleted sentence.

  1. Fig.4 description. COL20-1 – there is no symbol.

AU: Added (now Figure 5)

  1. I am curious why authors focus on lysine, due to there were no differences between COL10 and COL 20%. Why Authors did not focus also on others aa. Did they perform the correlation analysis (table5) for others aa?

AU: Yes correlation analysis was run on all AA.  Since including this information would make the table excessively large for a publication, we state that only glutamine and lysine showed any relationships. This is better clarified in the text of the article.

  1. Table 5 and lines 470-476 – there is no explanation for which animal (10, 20%COL or both) the correlation analysis was performed.

AU: Correlation analysis was done across both groups of animals (all 14, and not divided by group).  This was clarified in the text.

  1. Table 5 - why the Authors did not also perform the analysis of the relationship for the parenchymal area (indicated in Fig. 1).

AU: Correlation analysis was done with percent parenchymal epithelial area (PEA), label was modified in table to better clarify.

  1. Table 5, description of the results, Discussion - According to my knowledge glutamate is not the same as glutamine – please check the text, or provide the information about the glutamate-glutamine cycle.

AU: Yes, and we see that the discussion was not clear.  We modified discussion to clarify glutamine/glutamate and interpretation of glutamate’s negative relationship with parenchymal epithelial area.

  1. Discussion lines 521 - why the Authors stated that higher lysine levels favored a greater degree of a stem-progenitor cell? In the results section, there are no differences in lysine concentration in plasma of animals fed 10 and 20% COL. As was mentioned above, it cannot be determined as the degree of stem-progenitor cells.

AU: as above we remove reference to stem/progenitor cells, and noted it is currently unknown what is driving variation in 24h lysine and growth relationships.

Reviewer 2 Report

The proposed research aimed to indicate the relationship between level of colostrum intake and 24 h level of circulating amino acids, glucose and insulin with mammary parenchyma histological features, cell division and protein synthesis over the first week postnatal. The paper does contribute novelty information to a studied field and could be interesting to the readership of the journal. However in my opinion, for refine manuscript, Authors should make some additions:

  1. In the chapter Materials and methods:                                                           - Provide information about origin of evaluated piglets. Whether piglets were purebred or crossbred? If they came from prolific sows that information will be important.                                                                          - Provide litter size which gilt were taken from and which of sows lactation.                                                                                                             - During gestation, fetuses live at a uterine temperature that ranges from 38-40°C. However, at birth neonates suffer a drastic environmental change as they are exposed to an low temperature which makes them more vulnerable to stress-induced by cold. And we know that temperature for piglet must be higher than sows but explain why you maintained gilt at 24 h with 40°C on a nursery area.
  2. The reference to Table 3 in the text of Results chapter was omitted.
  3. I also suggest Authors add afollowing sources of bibliography:         Gourley K.M. et al. (2020) Effects of increased lysine and energy feeding duration prior to parturition on sow and litter performance, piglet survival, and colostrum quality https://doi.org/10.1093/jas/skaa105       Farmer (2018) Nutritional impact on mammary development in pigs: a review. https://doi.org/10.1093/jas/sky243                                         Garrison et al. (2017) Got colostrum? Effect of diet and feeding level on piglet colostrum intake and piglet quality https://doi.org/10.2527/asasmw.2017.12.236

Author Response

Response to review of: Manuscript ID: animals-1306258

Title: Plasma lysine concentration at 24 h after birth relates to mammary development in gilts at one week postnatal.

Dear Reviewers,

Thank you for taking the time to read and critique our article. Responding to your concerns improved the quality of the manuscript.

Please find below point-by-point response to your comments

Best, Theresa

Comments and Suggestions for Authors

Reviewer 2.

The proposed research aimed to indicate the relationship between level of colostrum intake and 24 h level of circulating amino acids, glucose and insulin with mammary parenchyma histological features, cell division and protein synthesis over the first week postnatal. The paper does contribute novelty information to a studied field and could be interesting to the readership of the journal. However in my opinion, for refine manuscript, Authors should make some additions:

  1. In the chapter Materials and methods:                                                           - Provide information about origin of evaluated piglets. Whether piglets were purebred or crossbred? If they came from prolific sows that information will be important.      

AU: The breed of sows (York X Landrace) and boar sire (Duroc) were added to M&M

  1. Provide litter size which gilt were taken from and which of sows lactation.       

AU: Birth litter size was added, and nursing litter size was standardized to 12–14 piglets per sow                                                                                                     

  1. During gestation, fetuses live at a uterine temperature that ranges from 38-40°C. However, at birth neonates suffer a drastic environmental change as they are exposed to an low temperature which makes them more vulnerable to stress-induced by cold. And we know that temperature for piglet must be higher than sows but explain why you maintained gilt at 24 h with 40°C on a nursery area.

AU: Temperature was chosen to prevent any risk of hyperthermia at this very young age, and information added to manuscript.

  1. The reference to Table 3 in the text of Results chapter was omitted.

AU: added

  1. I also suggest Authors add a following sources of bibliography.

AU: Thank you for these suggestions, below is information as to where these references were added to the manuscript.   

  1. Gourley K.M. et al. (2020) Effects of increased lysine and energy feeding duration prior to parturition on sow and litter performance, piglet survival, and colostrum quality https://doi.org/10.1093/jas/skaa105

AU: Added to discussion section on lysine, paragraph 4 of discussion.

  1. Farmer (2018) Nutritional impact on mammary development in pigs: a review. https://doi.org/10.1093/jas/sky243    

AU: Added to discussion section last sentence paragraph 3.

  1. Garrison et al. (2017) Got colostrum? Effect of diet and feeding level on piglet colostrum intake and piglet quality https://doi.org/10.2527/asasmw.2017.12.236

AU: Added to discussion section paragraph 4.

Round 2

Reviewer 1 Report

The authors highly improved the scientific level of the manuscript and in my opinion is much better now, however, I still found few stumbles which need to be clarified in the text.

In the text of the manuscript, the Authors use alternately “f” and “F” as the abbreviation of DNA or/and protein fractional synthesis. I recommend unifying the abbreviation in the text of the manuscript and in Tables 4 and 5, due to clarification.

For the first time, the Authors used the abbreviation “PEA” as the parenchymal epithelial area in line 222 of a revised version of the manuscript. In the subsequent parts of the manuscript, authors should use this abbreviation, not the whole name, or re-explain the meanings of this abbreviation. Please check also for others abbreviations, like “FSR” and “f”.

 Line 200,  225 – it should be °C not °C.

In the results section, the title and description of point 3.1 are missing.

Author Response

Response to Review2

  1. In the text of the manuscript, the Authors use alternately “f” and “F” as the abbreviation of DNA or/and protein fractional synthesis. I recommend unifying the abbreviation in the text of the manuscript and in Tables 4 and 5, due to clarification.

AU: edited throughout, f is used to indicate fraction of newly synthesize DNA or protein

  1. For the first time, the Authors used the abbreviation “PEA” as the parenchymal epithelial area in line 222 of a revised version of the manuscript. In the subsequent parts of the manuscript, authors should use this abbreviation, not the whole name, or re-explain the meanings of this abbreviation. Please check also for others abbreviations, like “FSR” and “f”.

AU: Edited to just PEA throughout.

  1. Line 200,  225 – it should be °C not °C.

AU: Edited

  1. In the results section, the title and description of point 3.1 are missing.

AU: Added